# Herb–Drug Interaction of Red Ginseng Extract and Ginsenoside Rc with Valsartan in Rats

**DOI:** 10.3390/molecules25030622

**Published:** 2020-01-31

**Authors:** Ji-Hyeon Jeon, Sowon Lee, Wonpyo Lee, Sojeong Jin, Mihwa Kwon, Chul Hwi Shin, Min-Koo Choi, Im-Sook Song

**Affiliations:** 1College of Pharmacy and Research Institute of Pharmaceutical Sciences, Kyungpook National University, Daegu 41566, Korea; kei7016@naver.com (J.-H.J.); okjin917@hanmail.net (S.L.); mihwa_k@naver.com (M.K.); tusshinn@gmail.com (C.H.S.); 2College of Pharmacy, Dankook University, Cheon-an 31116, Korea; woopyo906@gmail.com (W.L.); astraea327@naver.com (S.J.)

**Keywords:** red ginseng extract (RGE), ginsenoside Rc, herb–drug interaction, organic anion transporting polypeptide (Oatp), valsartan

## Abstract

The purpose of this study was to investigate the herb–drug interactions involving red ginseng extract (RGE) or ginsenoside Rc with valsartan, a substrate for organic anion transporting polypeptide (OATP/Oatp) transporters. In HEK293 cells overexpressing drug transporters, the protopanaxadiol (PPD)-type ginsenosides- Rb1, Rb2, Rc, Rd, Rg3, compound K, and Rh2-inhibited human OATP1B1 and OATP1B3 transporters (IC_50_ values of 7.99–68.2 µM for OATP1B1; 1.36–30.8 µM for OATP1B3), suggesting the herb–drug interaction of PPD-type ginsenosides involving OATPs. Protopanaxatriol (PPT)-type ginsenosides-Re, Rg1, and Rh1-did not inhibit OATP1B1 and OATP1B3 and all ginsenosides tested didn’t inhibit OCT and OAT transporters. However, in rats, neither RGE nor Rc, a potent OATP inhibitor among PPD-type ginsenoside, changed in vivo pharmacokinetics of valsartan following repeated oral administration of RGE (1.5 g/kg/day for 7 days) or repeated intravenous injection of Rc (3 mg/kg for 5 days). The lack of in vivo herb–drug interaction between orally administered RGE and valsartan could be attributed to the low plasma concentration of PPD-type ginsenosides (5.3–48.4 nM). Even high plasma concentration of Rc did not effectively alter the pharmacokinetics of valsartan because of high protein binding and the limited liver distribution of Rc. The results, in conclusion, would provide useful information for herb–drug interaction between RGE or PPD-type ginsenosides and Oatp substrate drugs.

## 1. Introduction

Ginseng is one of the most popular plants in Asia, Europe, and USA [1,2] owing to its vitality restoration and immunostimulating effect [1]. The therapeutic benefits of ginseng include anti-diabetic and anti-inflammatory effect and anti-oxidative response on chronic liver disease [3,4,5,6,7,8]. Ginseng is also commonly used due to its potential as a chemo-preventive agent and adjuvant therapy [9]. These pharmacological activities have also been observed for various ginsenosides mainly present in ginseng products [4].

Due to the growing use of herbal medicine and convenience of taking herbal formulations, herb–drug interactions caused by the co-administration of herbal medicine with therapeutic drugs have also rapidly increased from 13.8% in 2010 to 17.3% in 2013 among adverse drug reactions in China [10].

The most frequently reported cases of herb–drug interactions include the modulation of drug metabolizing enzymes and transporters by herbal medicines and the causative pharmacokinetic alterations of co-administered therapeutic drugs that may acts as substrates for drug metabolizing enzymes and transporters [8,11]. In case of ginseng interactions, it was reported that no herb–drug interaction between single oral dose of Korean red ginseng extract (RGE) (0.5–2.0 g/kg) and the probe substrates for five cytochrome P450 (CYP) enzymes (i.e., CYP1A2, 2C9, 2C19, 2D6, 3A) in mouse [12]. Repeated oral administration of RGE (0.5 g/kg for 2 weeks in mice and 85 mg total ginsenosides for 2 weeks in human) did not alter the metabolic activity of above 5 CYP enzymes in the mouse liver [13] and could not induce clinically significant interaction in human [14,15]. In a study conducted by Malati et al. [16], Korean ginseng (0.5 g capsule twice daily for 28 days) induced CYP3A activity and decrease plasma concentration of midazolam following oral administration of 8 mg midazolam. The expression levels of organic anion transporter 1 (Oat1) and Oat3 in the kidney and P-glycoprotein (P-gp) in the liver were increased by the repeated administration of RGE (30–300 mg/kg for 2 weeks) in mice, which were accompanied with the dose dependent decrease in the area under the plasma concentration-time curve (AUC) of fexofenadine, a substrate for P-gp [17]. In another study using rats, the bioavailability of fexofenadine was decreased by 16.1% following repeated administration of ginseng radix extract (150 mg/kg/day for 2 weeks), which may be explained by reduced absorption of fexofenadine due to the induction of intestinal P-gp [18]. Repeated RGE treatment was reported to decrease multidrug resistance-related protein 2 (Mrp2) mRNA and protein expression, consequently decreasing the biliary excretion of methotrexate and increasing plasma concentration [8]. Intestinal and hepatic organic cation transporter 1 (Oct1) expression was increased and decreased, respectively, in rats following repeated administration of RGE (1.5 g/kg for 7 days) [19]. Although repeated administration of RGE suggested modulation of transporter activity, the systematic pharmacokinetic ginseng-drug interaction on drug transporters and clinical evidence is still limited [20].

In addition to ginseng products, individual ginsenosides can also modulate drug-metabolizing enzymes or transporters. For example, Rb1, the most abundant ginsenoside in RGE, was found to significantly inhibit CYP2C9 (IC_50_ value of 2.4 µM), UDP-glucuronosyltransferase (UGT) 1A9 (IC_50_ value of 21.3 µM), organic anion transporting polypeptide 1B1 (OATP1B1) (IC_50_ value of 33.2 µM), and OATP1B3 (IC_50_ value of 4.8 µM). Other CYP enzymes, UGT enzymes, and transporters were not affected [14]. Ginsenosides could be grouped as protopanaxadiol (PPD)-type ginsenosides and protopanaxatriol (PPT)-type ginsenosides based on their hydroxylation site and their structure effected differentially on the UGT1A9 metabolic activity. F2, Rb1, Rb2, Rc, Rd, and Rg3, PPD-type ginsenosides, inhibited UGT1A9 activity with IC_50_ values ranging from 6.3 µM to 44.0 µM but PPT-type ginsenosides such as F1, Re, Rf, and Rg1 did not inhibit UGT1A9 [21]. The PPD-type ginsenoside Rg3 inhibited metabolic activities of UGT1A3 (IC_50_ value of 20.9 µM), UGT1A9 (IC_50_ value of 15.1 µM), and UGT2B7 (IC_50_ value of 23.1 µM). The PPD-type ginsenoside Rh2 has been reported to inhibit UGT1A3 with an IC_50_ value of 37.9 µM [22]. The PPD-type ginsenosides Rb1, Rc, and Rd inhibited OATP1B1 and OATP1B3 with IC_50_ values ranging from 0.2 µM to 4.6 µM. The PPT-type ginsenosides Rg1 and Re also inhibited OATP1B1 and OATP1B3 with IC_50_ values ranging from 39.4 µM to 133 µM [23]. In contrast to the reports of the inhibitory effect of ginsenosides on drug-metabolizing enzymes and transporters, little information is available on the in vivo pharmacokinetic ginsenosides-drug interactions. In addition, the inhibitory effect of PPD-type and PPT-type ginsenosides on drug transport activity has not been studied extensively. Considering the growing evidence of herb–drug interactions involving drug transporters [10], the aim of this study was to investigate the effect of RGE and individual ginsenoside (PPD-type as well as PPT-type) on drug transporters using in vitro cell system and/or in vivo animal model. Specifically, we focused on uptake transporters such as OCTs, OATs, and OATPs as these transporters regulate the tissue distribution, elimination, and pharmacokinetics of natural herbs and xenobiotics [10].

## 2. Results

### 2.1. Inhibitory Effect of Ginsenosides on Drug Transporters

The inhibitory effects of ginsenosides on drug transporters were evaluated using HEK293 cells overexpressing OCT1, OCT2, OAT1, OAT3, OATP1B1, OATP1B3 and HEK293-mock cells. First, we measured the uptake of probe substrates into respective transporters in the presence of typical inhibitors of the transporters for the system validation (Figure 1). Triethylammonium (TEA) inhibited OCT1 and OCT2 with IC_50_ values of 1177 μM and 1396 μM, respectively. Probenecid inhibited OAT1 and OAT3 with IC_50_ values of 2.33 μM and 1.49 μM, respectively. Rifampin inhibited OATP1B1 and OATP1B3 with IC_50_ values of 28.6 μM and 0.81 μM, respectively. The results were comparable with IC_50_ values of TEA reported in the literature (i.e., 1.4–7.4 mM for OCT1 and 2.05 mM for OCT2; 7.6 μM for OAT1 and 4.1 μM for OAT3; 0.8–22.8 μM for OATP1B1 and 0.8–6.4 μM for OATP1B3 [24,25,26,27,28].

We then evaluated the modulation of drug transporters by ginsenosides. We found that ginsenosides selectively inhibited OATP transport activities but not OCTs and OATs. Ginsenosides Rb1, Rb2, Rc, Rd, Rg3, compound K, Rh2, PPD, PPT, and Rh1 had IC_50_ values ranging from 1.36 μM to >100 μM. Most PPD-type ginsenosides inhibited both OATP1B1 and OATP1B3. Among them, tri-glycosylated PPD-type ginsenosides such as Rb1, Rb2, and compound K inhibited OATP1B3 with higher affinity (i.e., smaller IC_50_ values) than those of OATP1B1. However, most PPT-type ginsenosides did not inhibit OATP1B1 and OATP1B3 except for PPT (Figure 2 and Figure 3; Table 1). Contrary to the results on OATP inhibition, all 12 ginsenosides tested did not significantly inhibit OCT1, OCT2, OAT1, and OAT3 transporters (Table 1).

### 2.2. Valsartan as a Substrate for Oatp Transporter

Based on the significant inhibitory effect of ginsenosides on OATP1B1 and OATP1B3, we further evaluated the in vivo herb–drug interaction using a substrate drug for both OATP1B1 and OATP1B3. Valsartan was selected as a substrate for both OATP1B1 and OATP1B3 (Figure 4). Valsartan uptake by HEK293 cells expressing OATP1B1 and OATP1B3 was increased by 19.8-fold and 26.1-fold, respectively, compared with the uptake by HEK293-mock cells. OATP1B1- and OATP1B3-mediated valsartan uptake was inhibited by rifampin in a concentration-dependent manner, and the inhibition profile showed IC_50_ values of 13.8 μM and 3.6 μM, respectively. The results suggested that valsartan is a substrate for OATP1B1 and OATP1B3 transporters and OATP-mediated valsartan uptake was inhibited by the presence of representative OATP inhibitor, rifampin.

In addition to this, valsartan has been reported to be mainly eliminated via biliary excretion mediated by OATP1B1 and OATP1B3 in human and Oatps in rats. The contribution of Oatp transporters in the hepato-biliary excretion was about 70–85% in both rats and human [29,30]. The results suggest that valsartan could be used as a model drug for investigating OATP (in human) or Oatp (in rats)-mediated herb–drug interaction between valsartan and RGE or ginsenosides.

### 2.3. Effect of RGE on the Pharmacokinetics of Valsartan in Rats

We initially investigated the effect of rifampin on the valsartan pharmacokinetics following the intravenous injection of valsartan at a dose of 1 mg/kg as a positive control group and the results were shown in Figure 5A and Table 2.

The plasma concentration of valsartan was increased by co-treatment with rifampin, a typical inhibitor of OATP or Oatp transporters. Thus, pharmacokinetic parameters such as the area under the plasma concentration-time curve (AUC_24h_ and AUC_∞_) values were significantly higher than those of the control group. The clearance (CL) and volume of distribution (V_d_) of valsartan were decreased by rifampin co-administration. Taken together, rifampin inhibited OATP transport activity in vivo, thus decreasing the hepatic elimination of valsartan and increasing the plasma concentration of this drug. However, compared with the control group, repeated administration of RGE (1.5 g/kg/day for 7 days) did not affect the plasma concentration and pharmacokinetic parameters of valsartan (Figure 5B and Table 2). The results suggest that repeated RGE treatment did not inhibited Oatp transport activity in rats.

To explain the lack of herb–drug interaction between RGE and valsartan, we measured the plasma concentrations of ginsenosides following repeated administration of RGE using the previously developed analytical method by LC-MS/MS [19,31]. Among the 14 ginsenosides examined (Rb1, Rb2, Rc, Rd, Rh2, Rg3, F2, compound K, PPD, Re, Rh1, Rg1, F1, and PPT), 6 ginsenosides were detected in the plasma samples and the plasma concentrations of the 6 ginsenosides are shown in Figure 6. The plasma concentrations of the ginsenosides Rb1, Rb2, Rc, and Rd in rats after multiple administration of RGE (1.5 g/kg/day) for 1 week were consistent with previous results [8,19]. The ginsenosides Rh2, Rg3, F2, and compound K (intermediate metabolites of PPD-type ginsenosides [32], were not detected. PPD, a final metabolite of PPD-type ginsenosides, was detected in the rat plasma and showed a slow elimination process (Figure 6E and Table 3). Similarly, Re, Rh1, Rg1, and F1 (PPT-type ginsenosides and their intermediate metabolites [32]) were not detected in the rat plasma. PPT, a final metabolite of PPT-type ginsenosides, was detected and also showed a slow elimination process (Figure 6F and Table 3).

As shown in Table 3. the maximum plasma concentrations of Rb1, Rb2, Rc, Rd, PPD, and PPT were 17.84 ± 2.34 ng/mL (15.8 nM), 10.02 ± 1.04 ng/mL (9.1 nM), 11.67 ± 2.49 (10.6 nM), 5.10 ± 0.74 ng/mL (5.3 nM), 20.56 ± 9.47 ng/mL (48.4 nM), and 16.97 ± 8.99 ng/mL (38.5 nM), respectively. These concentrations might be far below the IC_50_ values required for the inhibition of Oatp transport activity and, therefore, the plasma PPD-type ginsenoside could not inhibit Oatp-mediated biliary excretion of valsartan effectively.

### 2.4. Effect of Ginsenoside Rc on the Pharmacokinetics of Valsartan in Rats

We further investigated herb–drug interaction between valsartan and individual ginsenoside. At first, the inhibitory effect of ginsenoside Rb1, Rb2, and Rc on the OATP1B1 and OATP1B3-mediated valsartan uptake was measured. Ginsenoside Rb1, Rb2, and Rc was selected considering its stability and high plasma concentation in rat plasma (based on Figure 6) and in human plasma [31,33] as well as its low IC_50_ value for OATP1B3 inhibition (2.28 μM, 1.76 μM, and 1.36 μM, respectively, Figure 3). As shown in Figure 7, Rb1, Rb2, and Rc inhibited both OATP1B1 and OATP1B3-mediated valsartan uptake in a concentration dependent manner and yielded IC_50_ values of 8.8–24.1 μM for OATP1B1 and 1.9–5.1 μM for OATP1B3. The results higher affinity of Rb1, Rb2, and Rc to OATP1B3 than OATP1B1 and the lowest IC_50_ value was shown in Rc inhibition on OATP1B3-mediated uptake of valsartan was consistent with the previous results (Figure 2 and Figure 3).

Next, we investigated whether the high plasma concentration of individual ginsenoside (Rc) above the IC_50_ value required for the inhibition of OATP/Oatp transporters could induce the in vivo herb–drug interactions. To achieve the highest and stable plasma concentration of Rc, it was injected intravenously for 5 days before the administration of valsartan. As shown in Figure 8B, the Rc concentration ranged from 7.8 μM to 34.1 μM. However, the plasma concentration of valsartan was not affected by Rc treatment (Figure 8A), and all pharmacokinetic parameters were not statistically different between the two groups (control group vs. Rc group) (Table 4). The results suggest that Rc did not inhibit the hepatic elimination of valsartan mediated by Oatp transporters in vivo even though the plasma concentration of Rc was greater than the IC_50_ value of Rc required for Oatp transport activity inhibition.

To investigate the cause of the minimal herb–drug interaction between Rc and valsartan, we measured the plasma and liver distribution of Rc following intravenous injection of Rc. As shown in Figure 9A, Rc was not widely distributed to the liver; thus, the liver concentration of Rc was lower than the plasma concentration of Rc and the liver-to-plasma concentration ratio of Rc was in the range of 0.13–0.2. In addition, these tri-glycosylated ginsenosides showed high protein binding in rat plasma and liver homogenates (Figure 9B). When calculated free Rc concentration in our system, free Rc concentration was estimated to be 0.08–0.34 μM in the rat plasma and 0.07–0.14 μM in the rat liver. As Oatp transporters are located in the sinusoidal membrane of hepatocytes, the low hepatic distribution and high protein binding of Rc may contribute to the negative inhibitory effect of Rc on Oatp transporters in vivo, which might result in the negligible pharmacokinetic interaction between Rc and valsartan. Similarly, limited herb–drug interaction between valsartan and ginsenoside Rb1 and Rb2 would be expected based on their similarity in the structure, protein binding features, and inhibitory effect on OATP transporters (Figure 7 and Figure 9B, Table 1).

## 3. Discussion

In the present study, the inhibitory effect of 12 ginsenosides (Rb1, Rb2, Rc, Rd, compound K, Rg3, Rh2, PPD, PPT, Re, Rg1, and Rh1) on HEK293 cells overexpressing drug transporters such as OCT1, OCT2, OAT1, OAT3, OATP1B1, and OATP1B3 was evaluated. The transport activity of OCT1, OCT2, OAT1, and OAT3 was not modulated by the 12 ginsenosides. However, Rb1, Rb2, Rc, Rd, Rg3, compound K, and Rh2 (PPD-type ginsenosides) inhibited OATP1B1 with IC_50_ values of 7.99–68.2 μM. PPD-type ginsenosides also inhibited OATP1B3 transport activity with higher affinity. The IC_50_ values of PPD-type ginsenosides for OATP1B3 inhibition ranged from 1.36 μM to 30.8 μM. On the other hand, PPT-type ginsenosides such as Re, Rg1, Rh1, and PPT did not inhibit OATP1B1 transport activity, and PPT and Rh1 inhibited OATP1B3 with IC_50_ values of 20.3 μM and >100 μM, respectively. These results suggest that PPD-type ginsenosides might induce herb–drug interactions via OATP1B1 and OATP1B3 inhibition.

This possible herb–drug interaction led us to investigate OATP-mediated in vivo pharmacokinetic herb–drug interactions because OATPs play important roles in the intestinal absorption and hepatic uptake of various therapeutic reagents. And a lot of clinically relevant herb–drug interactions and drug-drug interactions have been reported to be caused by the inhibition of OATPs [34]. For example, gemfibrozil and cyclosporine can increase the plasma exposure of pravastatin, pitavastatin, and atorvastatin through the inhibition of hepatic OATPs [34]. Quercetin (1500 mg/day for 7 days) and grapefruit juice (300 mL of pure juice) can inhibit intestinal OATP, thereby decreasing the absorption of talinolol [35]. Green tea and its marker component, epigallocatechin gallate, can also suppress the absorption of naldolol and rosuvastatin via the inhibition of intestinal OATP [35]. Given the importance of OATP (in human) and Oatp (in rat) in the pharmacokinetics and biliary excretion of valsartan [29,30], we monitored Oatp-mediated in vivo herb–drug interactions using valsartan as a substrate for Oatps in rats. The results revealed that the repeated administration of RGE and high dose of Rc did not significantly induce herb–drug interactions involving valsartan (Figure 5 and Figure 8).

Among PPD-type ginsenosides that inhibited in vitro OATP function, ginsenosides Rb1, Rb2, and Rc demonstrated high affinity for OATP1B3 inhibition (1.9–5.1 μM for OATP1B3; Figure 7). However, the maximum plasma concentrations (C_max_) of Rb1, Rb2, and Rc in rats were in the range of 5.3–15.8 nM following repeated administration of RGE (1.5 g/kg/day) for 7 days (Figure 6) and C_max_ of Rb1, Rb2, and Rc in human were 6.2–12.7 nM following repeated administration of RGE (3 g/day) for 14 days [31]. The selected RGE dose in this study is in the range of effective dose without significant toxicity and showed similar plasma concentrations of Rb1, Rb2, and Rc (5.3–15.8 nM in rats and 6.2–12.7 nM in human subjects) [19,33]. In numerous animal studies, the RGE dose has ranged from 200 mg/kg to 2.0 g/kg (i.e., 3–15 mg/kg of total ginsenosides) [36,37]. In human studies, RGE was administered to diabetic patients for 4 to 24 weeks at doses of 2.7–6.0g/day, which usually contained 50–100 mg ginsenosides/day [20,38]. The C_max_ values following oral administration of ginseng product in both rats and human would be far below the IC_50_ values required for OATP transport activity inhibition, which contribute to the limited herb–drug interaction between ginseng and OATP/Oatp substrates.

Co-administration of valsartan and Rc (3 mg/kg/day, iv for 5 days) resulted in the lack of herb–drug interaction between Rc and valsartan. The plasma concentration was ranged from 7.8 μM to 34.1 μM but unbound fraction of tri-glycosylated PPD-type ginsenosides (Rb1, Rb2, and Rc) was very low (0.1–0.2% in rat plasma, 0.4–0.5% in rat liver; Figure 9B). Moreover, the tri-glycosylated ginsenosides are hydrophilic and bulky and, thus, they are difficult to be readily distributed in the liver tissue. Taken together, high protein binding and limited liver distribution of tri-glycosylated PPD-type ginsenosides (Rb1, Rb2, and Rc) might contribute to the lack of in vivo pharmacokinetic herb–drug interactions involving valsartan in rats although their plasma concentration was maximized following repeated intravenous injection of single ginsenoside. Jiang et al. [23] reported that the unbound fraction of PPD-type ginsenosides was very low (0.4–0.9% in Rb1, Rc, and Rd) in the human plasma. Based on the similarity in the structure and protein binding features between rats and human and inhibitory effect on OATP transporters of Rb1, Rb2, and Rc, limited herb–drug interaction between valsartan and ginsenoside Rb1, Rb2, and Rc would be expected in human. In case of co-administration of valsartan and rifampin, a significant drug interaction between valsartan and rifampin was found (Figure 5A) because unbound concentration of rifampin (4.7–22.9 μM) would exceed the IC_50_ values required for OATP inhibition considering the plasma concentration (over 5 μg/mL for 12 h and C_max_ of 15.7–24.5 μg/mL) and protein binding of rifampin (23.1%) in rats following oral administration of rifampin 20 mg/kg [23,39,40].

Clinical herb–drug interactions between ginseng or ginsenosides and OATP1B1 or OATP1B3 have not been fully investigated. Nevertheless, in a previous study, single or repeated administration of red ginseng solution (>60% dried ginseng, three pouches/day once or for 2 weeks; equivalent to 85 mg total ginsenosides) did not have clinically significant inhibitory effects on the pharmacokinetics of pitavastatin, a selective substrate for OATP1B1 [14]. Furthermore, the clinically relevant pharmacokinetic ginseng or ginsenosides-valsartan interaction may not be caused based on the maximum concentrations of ginsenosides Rb1, Rb2, and Rc (6.2–12.7 nM) in human blood after repeated administration of red ginseng extract at high daily dose (3 g/day) [14,41]. In addition, 82-year-old male patient who took atorvastatin (80 mg), atenolol (50 mg), and aspirin (100 mg) reported drug-induced liver injury after concomitant ginseng intake. The patient’s symptoms regarding liver injury were resolved within 2 months after the cessation of both atorvastatin and the ginseng product [42]. After that, the case of liver injury might be deduced by the impaired elimination of atorvastatin through the inhibition of CYP3A4 and/or OATP1B1 activity by ginseng product [42]. However, based on the present results, the atorvastatin-induced liver injury in this patient may not be attributed to ginseng-atorvastatin interactions involving OATP1B1.

The benefits of ginseng and ginsenosides have been reported in cardiovascular diseases [2]. Ginseng is also widely used for individuals with cardiovascular risk factors such as hypertension and hypercholesterolemia [2]. The ginsenoside Rc has been found to have analgesic, anti-allergic, anti-tumor, and sedative effects [43]. The ginsenoside Rc may also be a strong anti-diabetic agent because it can markedly enhance glucose uptake [44]. Therefore, in conclusion, the findings showing the lack of herb–drug interactions between RGE or ginsenoside Rc and valsartan would provide useful information for patients taking anti-hypertensive, anti-diabetics, anti-tumor drugs such as valsartan and repaglinide that are substrate for OATPs transporters.

## 4. Materials and Methods

### 4.1. Materials

RGE was purchased from the Punggi Ginseng Cooperative Association (Youngjoo, Kyungpook, Korea). The ginsenosides Rb1, Rb2, Rc, Rd, Rg1, Rg3, Rh1, Rh2, compound K, Re, PPD, and PPT were purchased from the Ambo Institute (Daejeon, Korea). Berberine, caffeine, valsartan, probenecid, rifampin, TEA, sodium dodecyl sulfate (SDS), and Hank’s balanced salt solution (HBSS) were obtained from Sigma-Aldrich (St. Louis, MO, USA). Dulbecco’s modified Eagle’s medium (DMEM), fetal bovine serum (FBS), non-essential amino acids, and poly-D-lysine-coated 96-well plates were purchased from Corning Life Sciences (Woburn, MA, USA). [^3^H]Methyl-4-phenylpyridinium ([^3^H]MPP^+^; 2.9 TBq/mmol), [^3^H]para-aminohippuric acid ([^3^H]PAH; 0.13 TBq/mmol), [^3^H]estrone-3-sulfate ([^3^H]ES; 2.1 TBq/mmol), and [^3^H]estradiol-17β-d-glucuronide ([^3^H]EG; 2.2 TBq/mmol) were purchased from Perkin Elmer Inc. (Boston, MA, USA). Acetonitrile, methanol, and water were obtained from Fisher Scientific Co. (Fair Lawn, NJ, USA). All other chemicals and solvents were of reagent or analytical grade.

### 4.2. Inhibitory Effects of Ginsenosides on Drug Transporters

HEK293 cells overexpressing the OCT1, OCT2, OAT1, OAT3, OATP1B1, and OATP1B3 transporters (HEK293-OCT1, -OCT2, -OAT1, -OAT3, -OATP1B1, and -OATP1B3, respectively) and HEK293-mock cells (Corning Life Sciences; Woburn, MA, USA) were used and characterized as previously described [26,45,46].

HEK293 cells overexpressing drug transporters and HEK293-mock cells were seeded in poly-D-lysine-coated 96-well plates at a density of 10^5^ cells/well and were cultured in DMEM supplemented with 10% FBS, 5 mM non-essential amino acids, and 2 mM sodium butyrate in a humidified atmosphere with 8% CO_2_ at 37 °C. For the experiments, the growth medium was discarded after 24 h, and the attached cells were washed with HBSS and pre-incubated for 10 min in HBSS at 37 °C.

To examine the effects of ginsenosides and typical inhibitors on transporter activity, aliquots (100 μL) of HBSS containing the probe substrate and ginsenosides or typical inhibitors were added to the cells after aspirating pre-incubated HBSS. The concentrations and probe substrates were as follows: 0.1 μM [^3^H]MPP^+^ for OCT1 and OCT2, 0.1 μM [^3^H]PAH for OAT1, 0.1 μM [^3^H]ES for OAT3 and OATP1B1, and 0.1 μM [^3^H]EG for OATP1B3. The typical inhibitors were used as follows: TEA (0–50 mM) for OCT1 and OCT2, probenecid (0–250 μM) for OAT1 and OAT3, and rifampin (0–250 μM) for OATP1B1 and OATP1B3. The concentrations of the ginsenosides Rb1, Rb2, Rc, Rd, compound K, Rg3, Rh2, PPD, PPT, Re, Rg1, and Rh1 ranged from 0.1 µM to 100 μM. Then the uptake of probe substrates into HEK293-mock cells and HEK293-OCT1, -OCT2, -OAT1, -OAT3, -OATP1B1, and -OATP1B3 cells in the presence and absence of typical inhibitor or each ginsenoside with previously described concentration range was measured for 5 min. Immediately after placing the plates on ice, the cells were washed three times with 100 μL of ice-cold HBSS, followed by lysing with 50 μL of 10% SDS solution and mixing with 150 μL of Optiphase cocktail solution (Perkin Elmer Inc., Boston, MA, USA). The radioactivity of the probe substrates in the cells was determined using a liquid scintillation counter.

The uptake of valsartan 5 (μM)was measured for 5 min immediately after adding aliquots (100 μL) of HBSS containing 5 μM valsartan in the presence or absence of rifampin (1–100 μM) or tri-glycosylated PPD-type ginsenosides (Rb1, Rb2, and Rc; 0.1–100 μM) to the HEK293-mock cells and HEK293-OATP1B1 and –OAPT1B3 cells after aspirating pre-incubated HBSS. After 5 min incubation, the cells were washed three times with 100 μL of ice-cold HBSS, followed by lysing with 300 μL of 80% ice-cold methanol containing berberine 0.05 ng/mL and 0.1% formic acid for 15 min. After the centrifugation of cell lysate samples (16,000× *g*, 5 min, 4 °C), aliquots (4 μL) of cell lysate samples were injected into LC-MS/MS system for the analysis of valsartan. Transporter mediated uptake of probe substrates or valsartan was calculated by subtracting the uptake rate of probe substrates or valsartan into HEK293-mock cells from that into HEK293 cells expressing respective transporters.

### 4.3. Animals and Ethical Approval

Male Sprague-Dawley rats (6–7-weeks old, 220–250 g) were purchased from Samtako Co. (Osan, Korea). The animals were acclimatized for 1 week in an animal facility at Kyungpook National University. Food and water were provided ad libitum. All animal procedures were approved by the Animal Care and Use Committee of Kyungpook National University (Approval No. KNU 2017-21 and KNU 2019-83). To calculate and compare the pharmacokinetic parameters of valsartan and ginsenosides, we performed repeated blood sampling through the retro-orbital puncture under isoflurane anesthesia. During the experimental procedure, rats did not suffer from any significant injury or infection.

### 4.4. Pharmacokinetic Study

The rats were randomly divided into the control and rifampin groups. The rifampin group was orally administered with rifampin solution (20 mg/2 mL/kg, dissolved in DMSO: saline = 2:8, *v*/*v*) and the control group received only the vehicle via oral gavage. Valsartan was injected intravenously to both groups via the tail vein at 1 mg/mL/kg (dissolved in DMSO: saline = 2:8, *v*/*v*). Blood samples were collected via the retro-orbital vein at 0.25, 0.5, 1, 2, 4, 8, and 24 h following valsartan dosing. After the centrifugation of blood samples (16,000× *g*, 10 min, 4 °C), aliquots (50 μL each) of plasma samples were stored at –80 °C until the analysis of valsartan.

The rats were randomly divided into the control and RGE groups. The RGE group received a RGE suspension (1.5 g/mL/kg/day, in distilled water) for 7 days via oral gavage. The control group received distilled water for 7 days via oral gavage. After 1 h following the last RGE treatment, valsartan was injected intravenously to both groups via the tail vein at 1 mg/mL/kg (dissolved in DMSO: saline = 2:8, *v*/*v*). Blood samples were collected via the retro-orbital vein at 0.25, 0.5, 1, 2, 4, 8, 24, 30, and 48 h after valsartan dosing. After centrifugation of the blood samples at 16,000× *g* for 10 min, aliquots (50 μL each) of plasma samples were stored at −80 °C until the analysis of ginsenosides and valsartan.

The rats were randomly divided into the control and Rc groups. The Rc group was injected with Rc solution (3 mg/mL/kg, dissolved in saline) intravenously via the tail vein for 5 consecutive days. The control group received saline (1 mL/kg) for 5 consecutive days via the tail vein. After 1 h following the last Rc treatment, valsartan was injected intravenously via the femoral vein at 1 mg/kg. Blood samples were collected via the retro-orbital vein at 0.17, 0.33, 0.67, 1.5, 2, 4, 8, 24, and 48 h after valsartan dosing. After centrifugation of the blood samples (16,000× *g*, 10 min, 4 °C), aliquots (50 μL each) of plasma samples were stored at −80 °C until the analysis of the ginsenoside Rc and valsartan.

The rats were injected with Rc solution (3 mg/mL/kg, dissolved in saline) via the tail vein. Blood samples were collected from the abdominal artery, and the liver tissue was immediately excised, gently washed with ice-cold saline, and weighed after the rats were euthanized at 2, 12, and 48 h after intravenous injection of Rc. The blood samples were centrifuged (16,000× *g*, 10 min, 4 °C) and the liver tissue samples were homogenized with four volumes of saline. Aliquots (50 μL each) of plasma and liver homogenates were stored at −80 °C until the analysis of the ginsenoside Rc.

The protein binding of Rb1, Rb2, and Rc (1 μM) in rat plasma and liver homogenate was determined using a rapid equilibrium dialysis kit (ThermoFisher Scientific Korea, Seoul, Korea) according to the manufacturer’s instructions. Briefly, 100 μL of rat plasma and 10% liver homogenate samples containing Rb1, Rb2, or Rc (1 μM) were added to the sample chamber of a semipermeable membrane (molecular weight cut-off 8000 Da) and 300 μL of HBSS was added to the outer buffer chamber. Four hours after incubation at 37 °C on a shaking incubator at 300 rpm, aliquots (50 μL) were collected from both the sample and buffer chambers and treated with equal volumes of fresh HBSS and plasma, respectively, to match the sample matrices. The matrix-matched samples (100 μL) were mixed with 300 μL of 80% ice-cold methanol containing berberine 0.05 ng/mL and 0.1% formic acid for 15 min. After centrifugation (16,100× *g*, 5 min, 4 ^o^C), an aliquot (4 μL) from the sample was injected into LC-MS/MS system.

Plasma protein binding was calculated using the following equation [47].
(1)$$Undiluted\;free\;drug\;fraction\;\left(fu\right)=\;\frac{Drug\;concentration\;in\;buffer\;chamber}{Drug\;concentration\;in\;plasma\;chamber}$$

Tissue protein binding was calculated using the following equations, and a dilution factor (D as a value of 10) was used since we used 10% liver homognenates [47,48].
(2)$$\mathrm{Diluted}\;\mathrm{free}\;\mathrm{drug}\;\mathrm{fraction}\;\left({\mathrm{fu}}^{\prime }\right)=\;\frac{\mathrm{Drug}\;\mathrm{concentration}\;\mathrm{in}\;\mathrm{buffer}\;\mathrm{chamber}}{\mathrm{Drug}\;\mathrm{concentration}\;\mathrm{in}\;\mathrm{liver}\;\mathrm{homogenate}\;\mathrm{chamber}}$$
(3)$$\mathrm{Undiluted}\;\mathrm{free}\;\mathrm{drug}\;\mathrm{fraction}\;\left(\mathrm{fu}\right)=\;\frac{1/D}{\left(\frac{1}{{\mathrm{fu}}^{\prime }}-1\right)+1/D}=\;\frac{0.1}{\left(\frac{1}{{\mathrm{fu}}^{\prime }}-1\right)+0.1}$$

### 4.5. LC-MS/MS Analysis of Valsartan

The concentration of valsartan was analyzed using a modified Liquid chromatography–mass spectrometry (LC-MS/MS) method as previously reported by Yamashiro et al. [29] using an Agilent 6470 Triple Quad LC–MS/MS system (Agilent, Wilmington, DE, USA).

Briefly, aliquots (50 μL) of plasma samples were mixed with 350 μL of an internal standard (IS) solution (berberine 0.05 ng/mL in methanol) and the mixtures were vortexed for 15 min. After centrifugation (16,100× *g*, 5 min, 4 °C), 100 μL of the supernatant was transferred to a clean tube, evaporated and an aliquot (4 μL) from the sample was injected into LC-MS/MS system.

The samples were eluted through a Synergy Polar RP column (2.0 mm × 150 mm, 4 µm particle size) (Phenomenex, Torrance, CA, USA) using a mobile phase consisting of methanol and water (75:25, *v*/*v*) with 0.1% formic acid at a flow rate of 0.2 mL/min. Valsartan and berberine (IS) were detected at a retention time (T_R_) of 2.6 min and 3.5 min, respectively, by electrospray ionization in the positive ion mode. Quantification was performed in the selected reaction-monitoring mode at *m*/*z* 436.1 → 291.0 for valsartan and *m*/*z* 336.1 → 320.0 for berberine. Plasma calibration standards for the measurement of valsartan ranged from 1 ng/mL to 2000 ng/mL, and the intraday and interday accuracy ranged from 93.35% to 99.63%. The intraday and interday precision ranged from 1.80% to 9.23%.

### 4.6. LC-MS/MS analysis of Ginsenosides

The concentration of compound K was analyzed using a modified LC-MS/MS method of Jin et al. [31] using an Agilent 6470 Triple Quad LC–MS/MS system (Agilent, Wilmington, DE, USA).

For the detection of Rb1, Rb2, Rc, Rd, Re, Rg1, Rh2, and Rg3, aliquots (50 μL) of plasma and liver homogenates were mixed with 350 μL of IS solution (0.05 ng/mL berberine in methanol) and vortexed for 15 min. After centrifugation (16,100× g for 5 min, 4 °C), 200 μL of the supernatant was transferred to a clean tube and evaporated. The residue was reconstituted with 100 μL of 70% methanol consisting of 0.1% formic acid. An aliquot (20 μL) from the sample was injected into the LC-MS/MS system. The samples were eluted through a Synergy Polar RP column (2.0 mm × 150 mm, 4 µm particle size) (Phenomenex, Torrance, CA, USA) with a gradient mobile phase consisting of 0.1% formic acid in water (phase A) and 0.1% formic acid in methanol (phase B) as follows: 69% of phase B for 0–2.0 min, 69–85% of phase B for 2.0–4.0 min, and 85–69% of phase B for 6.0–6.5 min at a flow rate of 0.27 mL/min. The ginsenosides Rb1, Rb2, Rc, Rd, Re, Rg1, Rh2, Rg3, and berberine (IS) were detected at *m*/*z* 1131.6 → 365.1 (for Rb1, T_R_ 4.6 min), *m*/*z* 1101.6 → 335.1 (for Rb2 and Rc, T_R_ 5.7 min and 4.8 min, respectively), *m*/*z* 969.9 → 789.5 (for Rd and Re, T_R_ 6.8 min and 2.1 min, respectively), *m*/*z* 824.0 → 643.6 (for Rg1, T_R_ 2.2 min), *m*/*z* 587.4 → 4.7.4 (for Rh2, T_R_ 10.7 min), *m*/*z* 807.5 → 365.1 (for Rg3, T_R_ 9.3 min), and *m*/*z* 336.1 → 320.0 (for berberine, T_R_ 4.5 min) in the positive ion mode.

For the detection of Rh1, compound K, PPD, and PPT, aliquots (50 μL) of plasma and liver homogenates were mixed with 50 μL of IS solution (25 ng/mL caffeine in water) and 600 μL of methyl tertiary-butyl ether (MTBE), vortexed for 15 min, and centrifuged at 16,100× g for 5 min. After freezing the aqueous layer at −80 °C for 2 h, the upper organic layer was transferred to a clean tube and evaporated to dryness. The residue was reconstituted with 150 μL of 85% methanol. An aliquot (20 μL) from the sample was injected into the LC-MS/MS system. The samples were eluted through an Omega Polar C18 column (2.1 mm × 100 mm, 3 µm particle size) (Phenomenex, Torrance, CA, USA) using a mobile phase consisting of 0.1% formic acid in water: 0.1% formic acid in methanol (15:85, *v*/*v*) at a flow rate of 0.2 mL/min. The ginsenosides Rh1, Rh2, compound K, PPD, PPT, and caffeine (IS) were detected at *m*/*z* 603.4 → 423.4 (for Rh1, T_R_ 2.7 min), *m*/*z* 645.5 → 203.1 (for compound K, T_R_ 5.3 min), *m*/*z* 425.3 → 109.1 (for PPD, T_R_ 7.7 min), *m*/*z* 441.3 → 109.1 (for PPT, T_R_ 3.5 min), and *m*/*z* 195 → 138 (for caffeine, T_R_ 3.6 min) in the positive ion mode. The calibration standards for the measurement of 12 ginsenosides ranged from 0.5 ng/mL to 200 ng/mL, and the coefficient of variance for intraday and interday accuracy and precision were less than 15%.

### 4.7. Data Analysis

In the inhibition studies, the uptake rate of substrate by HEK293 cells overexpressing the respective transporters was used as the control (100%) and the uptake rate of substrates in the presence of typical inhibitors or ginsenosides expressed as a percentage of the control. The inhibition data were fitted to an inhibitory effect model [26] using Sigma plot (version 10.0; Systat Software Inc., San Jose, CA, USA). IC_50_ value indicated the half-maximal inhibitory concentration of the inhibitor.

Pharmacokinetic parameters were calculated from plasma concentration-time profile using non-compartment analysis of WinNonlin (version 5.1; Pharsights, Cary, NC, USA). The statistical significance was assessed by t-test using Statistical Package for the Social Sciences (version 24.0; SPSS Inc., Chicago, IL, USA).

## Figures and Tables

**Figure 1 molecules-25-00622-f001:**
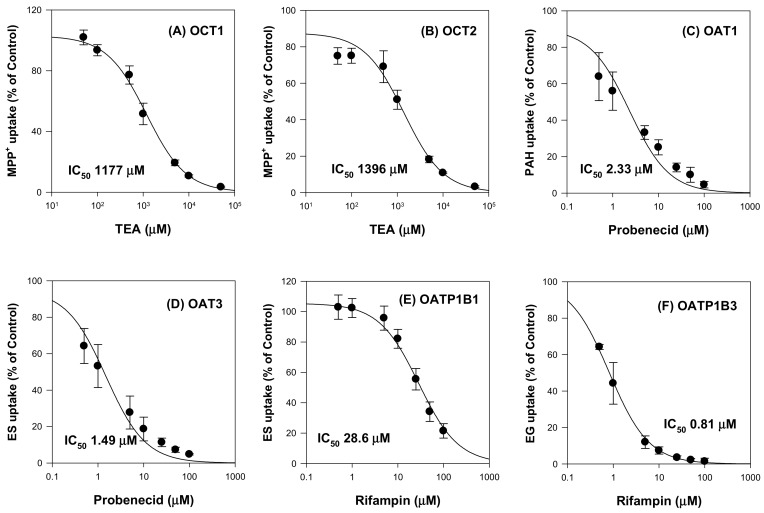
Inhibitory effect of typical inhibitors on the (**A**) OCT1, (**B**) OCT2, (**C**) OAT1, (**D**) OAT3, (**E**) OATP1B1, and (**F**) OATP1B3-mediated uptake. Transporter mediated uptake of probe substrate were calculated by subtracting the uptake in HEK293-mock cells from the uptake in HEK293 cells overexpressing respective transporters. The concentrations and probe substrates were as follows: 0.1 μM [^3^H]methyl-4-phenylpyridinium (MPP^+^) for OCT1 and OCT2, 0.1 μM [^3^H]para-aminohippuric acid (PAH) for OAT1, 0.1 μM [^3^H]estrone-3-sulfate (ES) for OAT3 and OATP1B1, and 0.1 μM [^3^H]estradiol-17β-d-glucuronide (EG) for OATP1B3. The typical inhibitors were used as follows: TEA (0–50 mM) for OCT1 and OCT2, probenecid (0–250 μM) for OAT1 and OAT3, and rifampin (0–250 μM) for OATP1B1 and OATP1B3. Data are the mean ± SD from triplicate measurements.

**Figure 2 molecules-25-00622-f002:**
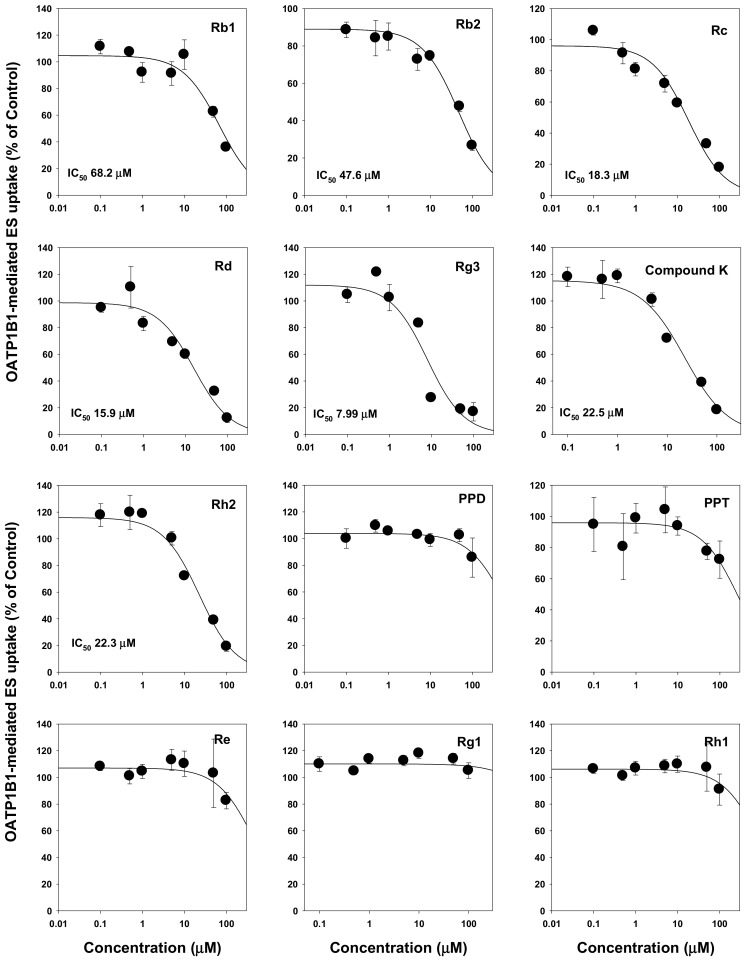
Inhibitory effect of Rb1, Rb2, Rc, Rd, compound K, Rg3, Rh2, PPD, PPT, Re, Rg1, and Rh1 on the OATP1B1-mediated transport of [^3^H]estrone-3-sulfate (ES). OATP1B1-mediated ES uptake were calculated by subtracting 0.1 μM [^3^H]ES uptake in HEK293-mock cells from 0.1 μM [^3^H]ES uptake in HEK293-OATP1B1 cells. Data are the mean ± SD from triplicate measurements.

**Figure 3 molecules-25-00622-f003:**
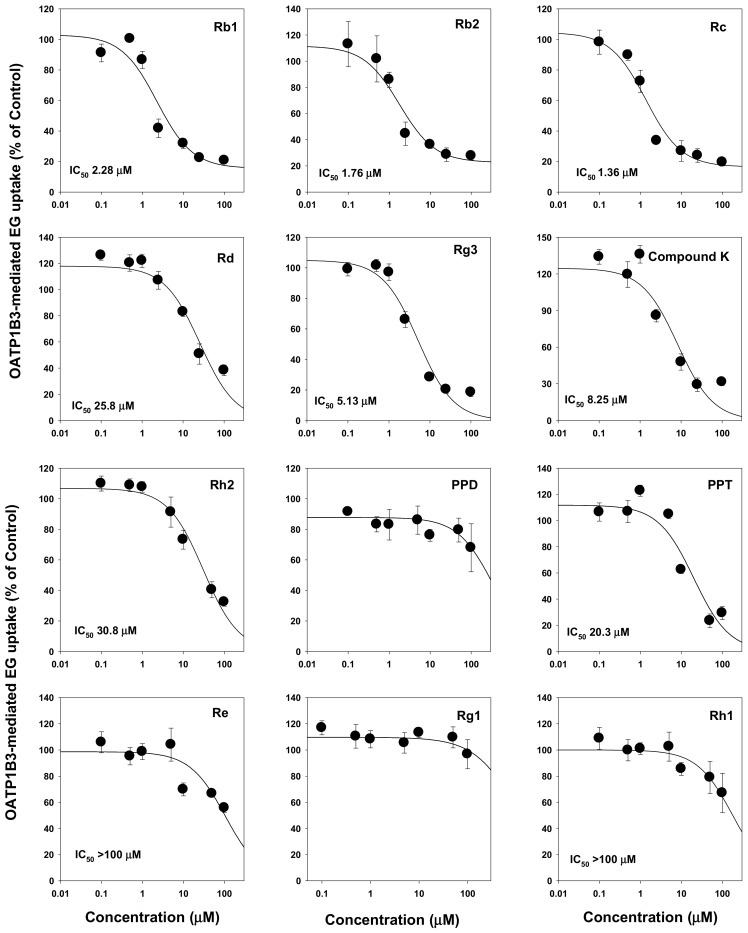
Inhibitory effect of Rb1, Rb2, Rc, Rd, compound K, Rg3, Rh2, PPD, PPT, Re, Rg1, and Rh1 on the OATP1B3-mediated uptake of [^3^H]estradiol-17β-D-glucuronide (EG). OATP1B3-mediated EG uptake were calculated by subtracting 0.1 μM [^3^H]EG uptake by HEK293-mock cells from 0.1 μM [^3^H]EG uptake by HEK293-OATP1B3 cells. Data are the mean ± SD from triplicate measurements.

**Figure 4 molecules-25-00622-f004:**
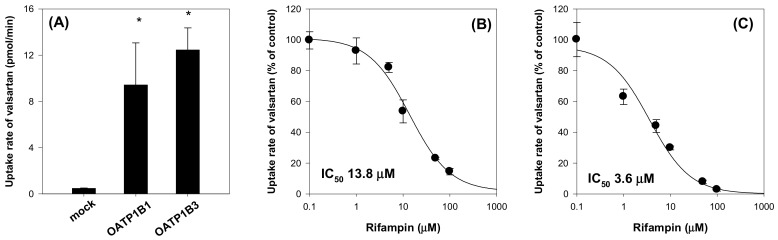
(**A**) Uptake of valsartan (5 μM) by HEK293-mock cells and HEK293 cells expressing OATP1B1 and OATP1B3. Inhibitory effect of rifampin on the (**B**) OATP1B1- and (**C**) OATP1B3-mediated uptake of valsartan. Data points represent the mean ± SD from triplicate measurements. **p* < 0.05 compared with HEK293-mock cells.

**Figure 5 molecules-25-00622-f005:**
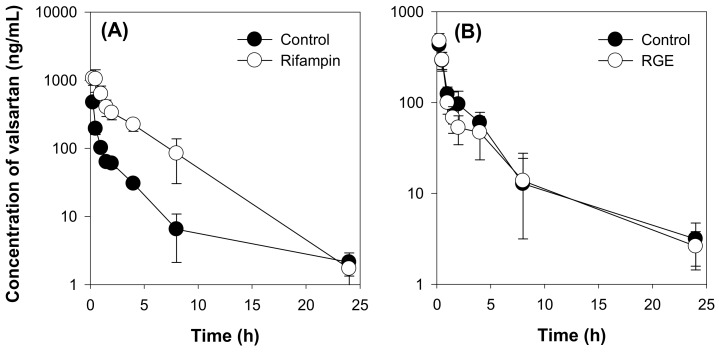
(**A**) Plasma concentration-time profile of valsartan in the control and rifampin (20 mg/kg) groups following intravenous injection of valsartan at a dose of 1 mg/kg in rats. (**B**) Plasma concentration-time profile of valsartan in the control and red ginseng extract (RGE, 1.5 g/kg/day for 7 days) groups following intravenous injection of valsartan at a dose of 1 mg/kg in rats. Data points represent the mean ± SD of four different rats per group.

**Figure 6 molecules-25-00622-f006:**
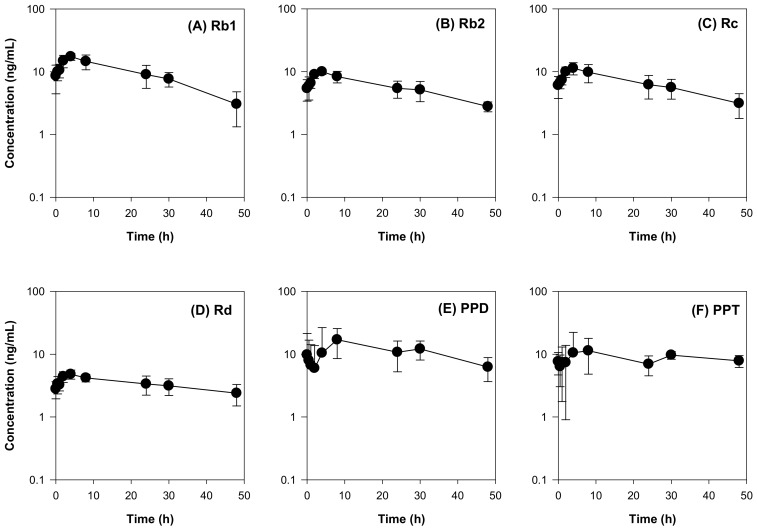
Plasma concentration-time profiles of the ginsenosides (**A**) Rb1, (**B**) Rb2, (**C**) Rc, (**D**) Rd, (**E**) PPD, and (**F**) PPT in the rat plasma after 1-week repeated administration of red ginseng extract (RGE). Data represent the mean ± SD of four rats.

**Figure 7 molecules-25-00622-f007:**
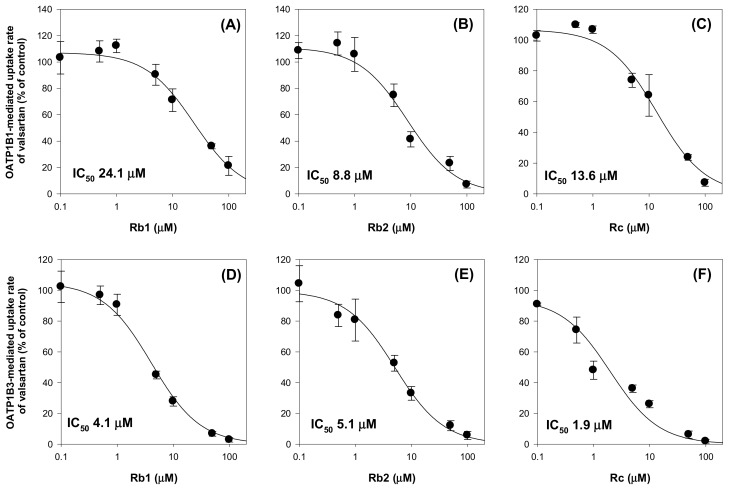
Inhibitory effect of Rb1 (**A**,**D**), Rb2 (**B**,**E**), and Rc (**C**,**F**) on the OATP1B1- and OATP1B3-mediated uptake of valsartan. OATP1B1- and OATP1B3-mediated valsartan uptake was calculated by subtracting valsartan uptake (5 μM) by HEK293-mock cells from valsartan uptake (5 μM) by HEK293-OATP1B1 and -OATP1B3 cells, respectively. Data points represent the mean ± SD from triplicate measurements.

**Figure 8 molecules-25-00622-f008:**
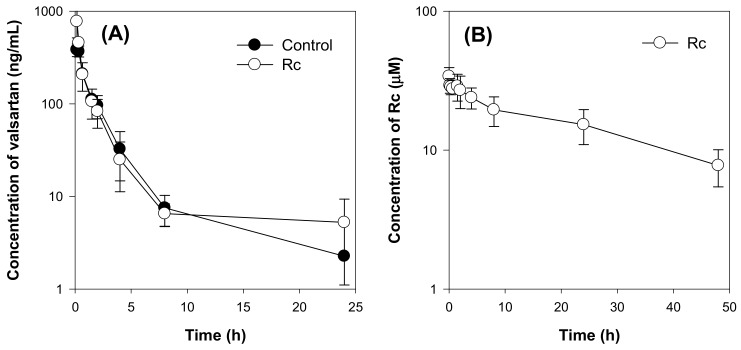
(**A**) Plasma concentration-time profile of valsartan in the control and Rc groups following intravenous injection of valsartan at a dose of 1 mg/kg. Rc was injected intravenously for 5 days at a dose of 3 mg/kg/day. (**B**) Plasma concentration-time profile of Rc in the Rc group. Data points represent the mean ± SD of three different rats per group.

**Figure 9 molecules-25-00622-f009:**
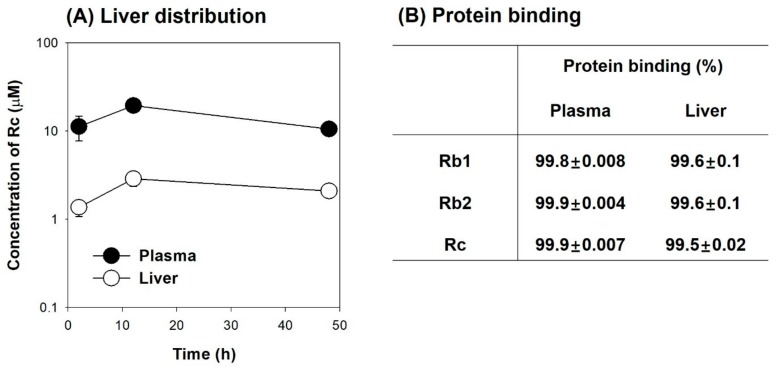
(**A**) Plasma (●) and liver (○) concentrations of Rc following intravenous injection of Rc at a dose of 3 mg/kg. (**B**) Protein binding of Rb1, Rb2, and Rc in the rat plasma and liver homogenates using a rapid equilibrium dialysis device. Data were expressed as the mean ± SD of three different rats or triplicated measurements.

**Table 1 molecules-25-00622-t001:** Inhibitory effect of ginsenosides on drug transporters.

	Ginsenosides	IC_50_ (μM)					
		OCT1	OCT2	OAT1	OAT3	OATP1B1	OATP1B3
PPD-type ginsenosides	Rb1	NI	NI	NI	NI	68.2	2.28
	Rb2	NI	NI	NI	NI	47.6	1.76
	Rc	NI	NI	NI	NI	18.3	1.36
	Rd	NI	NI	NI	NI	15.9	25.8
	Rg3	NI	NI	NI	NI	7.99	5.13
	Compound K	NI	NI	NI	NI	22.5	8.25
	Rh2	NI	NI	NI	NI	22.3	30.8
	PPD	NI	NI	NI	NI	NI	NI
PPT-type ginsenosides	PPT	NI	NI	NI	NI	NI	20.3
	Re	NI	NI	NI	NI	NI	>100
	Rg1	NI	NI	NI	NI	NI	NI
	Rh1	NI	NI	NI	NI	NI	>100

NI: No significant inhibition; >100: weak inhibition but IC_50_ value over 100 μM.

**Table 2 molecules-25-00622-t002:** Pharmacokinetic parameters of valsartan following intravenous injection of valsartan at a dose of 1 mg/kg in rats.

Treatment	Valsartan + Rifampin		Valsartan + RGE	
PK Parameters	Control	Rifampin	Control	RGE
T_1/2_ (h)	4.48 ± 1.24	2.79 ± 0.32 *	4.55 ± 0.98	4.59 ± 0.73
C_0_ (ng/mL)	1169.59 ± 118.78	1633.08 ± 1308.67	647.53 ± 50.49	781.47 ± 211.04
AUC_24h_ (ng·h/mL)	667.56 ± 219.43	3318.65 ± 809.25 *	859.36 ± 234.46	776.59 ± 228.82
AUC_∞_ (ng·h/mL)	681.03 ± 215.12	3325.83 ± 807.91 *	881.62 ± 247.34	794.48 ± 226.13
MRT (h)	2.99 ± 0.47	3.37 ± 0.83	3.76 ± 1.16	3.81 ± 0.05
CL (mL/min/kg)	25.96 ± 6.30	15.55 ± 5.69 *	20.03 ± 5.48	22.47 ± 7.12
Vd (mL/kg)	78.30 ± 25.66	55.03 ± 26.29 *	72.16 ± 15.37	85.38 ± 25.82

Data represent mean ± SD of four rats per group. * *p* < 0.05 compared with control group. T_1/2_: elimination half-life; C_0_: initial plasma concentration at 1 h; AUC_24h_ or AUC_∞_: Area under plasma concentration-time curve from zero to 24 h or infinity; MRT: mean residence time; CL: systemic clearance; Vd: Volume of distribution.

**Table 3 molecules-25-00622-t003:** Pharmacokinetic parameters of ginsenosides in the rat plasma after 1-week repeated administration of red ginseng extract (RGE).

Ginsenosides	Pharmacokinetic Parameters				
	AUC (ng∙h/mL)	C_max_ (ng/mL)	T_max_ (h)	MRT (h)	T_1/2_ (h)
Rb1	454.57 ± 111.33	17.84 ± 2.34	3.33 ± 1.15	17.13 ± 2.09	16.87 ± 5.81
Rb2	282.80 ± 58.90	10.02 ± 1.04	3.33 ± 1.15	18.61 ± 1.43	30.19 ± 5.48
Rc	320.75 ± 97.23	11.67 ± 2.49	3.33 ± 1.15	18.35 ± 1.89	24.24 ± 7.51
Rd	163.83 ± 39.07	5.10 ± 0.74	2.17 ± 1.76	20.84 ± 1.68	43.49 ± 17.54
PPD	542.01 ± 141.09	20.56 ± 9.47	16.50 ± 12.48	21.56 ± 3.31	26.95 ± 18.33
PPT	429.91 ± 105.75	16.97 ± 8.99	16.50 ± 12.48	23.12 ± 2.26	46.38 ± 13.27

Data represent mean ± SD of four rats per group. AUC: area under the plasma concentration-time curve from 0 to 48 h C_max_: maximum plasma concentration; T_max_: time to reach C_max_; MRT: Mean residence time; T_1/2_: Half-life.

**Table 4 molecules-25-00622-t004:** Pharmacokinetic parameters of valsartan and Rc following intravenous injection of valsartan at a dose of 1 mg/kg in rats.

Valsartan			Rc	
Parameters	Control	Rc Treatment	Parameters	Rc Treatment
T_1/2_ (h)	2.41 ± 1.30	3.30 ± 1.51	T_1/2_ (h)	27.51 ± 4.26
C_0_ (ng/mL)	400.66 ± 77.85	1451.25 ± 998.21	C_0_ (μM)	34.04 ± 5.22
AUC_24h_ (ng·h/mL)	639.41 ± 80.71	808.86 ± 111.90	AUC_24h_ (μM·h)	748.59 ± 184.79
AUC_∞_ (ng·h/mL)	657.34 ± 84.89	836.42 ± 112.34	AUC_∞_ (μM·h)	1064.50 ± 324.96
MRT (h)	2.27 ± 0.67	2.94 ± 1.52	MRT (h)	38.50 ± 5.11
CL (mL/h/kg)	25.66 ± 3.50	20.18 ± 2.86		
Vd (L/kg)	57.14 ± 13.24	57.65 ± 27.32		

Data represent mean ± SD of three rats per group T_1/2_: elimination half-life; C_0_: initial plasma concentration at 1 h; AUC_24h_ or AUC_∞_: Area under plasma concentration-time curve from zero to 24 h or infinity; MRT: mean residence time; CL: systemic clearance; Vd: Volume of distribution.

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
