# Peer review of "Herb–Drug Interaction of Red Ginseng Extract and Ginsenoside Rc with Valsartan in Rats"

_molecules, 2020, doi:10.3390/molecules25030622_

Round 1
Reviewer 1 Report
The authors present a well conducted in vitro and in vivo (experimental animals) study investigating the potential interaction between RGE and ginsenoside Rc with drug Transporters. While most of the methods are well described and executed, the manuscript suffers from one relevant weakness:
The authors in fact show concentration-dependent inhibition of substrate (e.g. MPP, PAH, etc.) uptake but it remains open whether this is truly mediated by the respective drug transporter overexpressed. That means the authors did not correct for unspecific substrate uptake. This Reviewer would suggest to conduct all experiments in HEK293-mock cells to estimate the contribution of unspecific uptake. Later on, the true drug transporter inhibition can be calculated as follows:
(Inhibited OCT cell / Uninhibited OCT cells) / (Inhibited mock cells / Uninhibited mock cells).
Author Response
Responses to Reviewer 1’s comments
The authors present a well conducted in vitro and in vivo (experimental animals) study investigating the potential interaction between RGE and ginsenoside Rc with drug Transporters. While most of the methods are well described and executed, the manuscript suffers from one relevant weakness:
The authors in fact show concentration-dependent inhibition of substrate (e.g. MPP, PAH, etc.) uptake but it remains open whether this is truly mediated by the respective drug transporter overexpressed. That means the authors did not correct for unspecific substrate uptake. This Reviewer would suggest to conduct all experiments in HEK293-mock cells to estimate the contribution of unspecific uptake. Later on, the true drug transporter inhibition can be calculated as follows:
(Inhibited OCT cell / Uninhibited OCT cells) / (Inhibited mock cells / Uninhibited mock cells).
Answer> Thank you for the valuable comments. According to the reviewer’s comments, we performed all experiments (uptake of probe substrates and valsartan in the presence or absence of typical inhibitors) in HEK293-mock cells and HEK293 cells overexpressing respective transporters. And net transporter-mediated uptake of substrates was calculated by subtracting the uptake of substrates into HEK293-mock cells from the uptake of substrates into HEK293 cells overexpressing respective transporters. All data were changed and the IC50 values were re-calculated using new data set during the revision. In the revised manuscript, Figures 1, 2, 3 and 4 and Table 1 were changed accordingly.

Reviewer 2 Report
The reviewer does not support the acceptance of this manuscript.
(1) The animal handling procedures are not acceptable. Repeated retro-orbital puncture is not acceptable for blood collection.
(2) The study is not clinical relevant. The plasma levels of ginsenosides are so low (figure 6) and they are unlikely cause interaction when judging from their IC50 levels.
Author Response
Responses to Reviewer 2’s comments
The reviewer does not support the acceptance of this manuscript.
(1) The animal handling procedures are not acceptable. Repeated retro-orbital puncture is not acceptable for blood collection.
Answer> As the reviewer pointed, we have 6-9 repeated retro-orbital puncture for 48 h for the pharmacokinetic study, which is not generally accepted. However, to calculate and compare the pharmacokinetic parameters we need repeated blood sampling and the retro-orbital puncture was performed under isoflurane anesthesia and therefore, rats did not suffer from any significant injury or infection. And our animal procedure was approved by the Animal Care and Use Committee of Kyungpook National University (Approval No. KNU 2017-21 and KNU 2019-83). We ask generous understanding on this issue.
(2) The study is not clinical relevant. The plasma levels of ginsenosides are so low (figure 6) and they are unlikely cause interaction when judging from their IC50 levels.
Answer> As the reviewer pointed, the lack of herb-drug interaction between red ginseng and valsartan following oral administration of red ginseng extract could be expected because of low plasma concentrations of ginsenosides. However, another purpose of this study was to investigate the possibility of herb-drug interaction between individual ginsenoside and valsartan because individual ginsenoside is under clinical trials for the development of co-therapy with anti-cancer, anti-diabetic, and anti-rheumatoid arthritis drugs. For this purpose we investigated the inhibitory potential of individual ginsenosides on drug transporters and in vivo pharmacokinetic study following co-administration of valsartan and Rc. From the study results, even high plasma concentrations of PPD-type ginsenoside would not cause significant herb-drug interaction with valsartan, a substrate for OATPs/Oatps and these results would also provide useful information for pharmacokinetic interaction between PPD-type ginsenosides and Oatp substrate drugs.

Reviewer 3 Report
The present manuscript describes findings from an in vitro and animal study on the inhibitory effects of ginsenosides on valsartan pharmacokinetics as a substrate of the uptake transporters OATP1B1 and OATP1B3. The experiments are conducted very systeamtically and are well controlled. The findings are very relevant to the uptake transporter and herb-drug interactions community.
I have just two minor suggestions:
The authors state in the introduction that no effects on CYP enzymes were found. They thereby ignore a relevant study by Malaty CY et al. J Clin Pharmacol. 2012, who show an induction of CYP3A4 in healthy volunteers. There is always discussion on the transferability of animal data on the human setting in the pharmacokinetics community. A clearer picture could have been drawn if the authors used HEK cells expressing rat OATPs in addition to human OATPs and found similar results. These experiments might have already been performed by others, but may then be more explicitly discussed in the present manuscript.Author Response
Responses to Reviewer 3’s comments
The present manuscript describes findings from an in vitro and animal study on the inhibitory effects of ginsenosides on valsartan pharmacokinetics as a substrate of the uptake transporters OATP1B1 and OATP1B3. The experiments are conducted very systeamtically and are well controlled. The findings are very relevant to the uptake transporter and herb-drug interactions community.
I have just two minor suggestions:
The authors state in the introduction that no effects on CYP enzymes were found. They thereby ignore a relevant study by Malaty CY et al. J Clin Pharmacol. 2012, who show an induction of CYP3A4 in healthy volunteers.Answer> According to the reviewer’s suggestion, we added the study results of Malaty et al. in the Introduction of the revised manuscript: (Page 2, line 51) In a study conducted by Malati et al. [16], Korean ginseng (0.5 g capsule twice daily for 28 days) induced CYP3A activity and decrease plasma concentration of midazolam following oral administration of 8 mg midazolam.
There is always discussion on the transferability of animal data on the human setting in the pharmacokinetics community. A clearer picture could have been drawn if the authors used HEK cells expressing rat OATPs in addition to human OATPs and found similar results. These experiments might have already been performed by others, but may then be more explicitly discussed in the present manuscript.
Answer> We totally agree with the reviewer’s comments but we did not have HEK cells expressing rat Oatp transporters to explore the in vitro inhibition study in rats. Moreover, we could not find any references that reported in vitro inhibitory effect of ginsenosides on rat Oatp transporters. However, Jiang et al. [Molecular mechanisms governing different pharmacokinetics of ginsenosides and potential for ginsenoside-perpetrated herb-drug interactions on OATP1B3. Br J Pharmacol 2015, 172, 1059-1073] investigated the human OATP and rat Oatp mediated transport features of ginsenosides. PPD-type ginsenosides (Rb1, Rc and Rd) were not substrates for Oatp/OATP transporters but PPT-type ginsenosides (Rg1, Re, and R1) were substrates for Oatp/OATP transporters with comparable intrinsic clearance between rat and human. Therefore, we can deduce the kinetic similarity between rat Oatps and human OATPs although these comparison should be performed in the future. Instead, we compared plasma concentration of ginsenosides in rat and human plasma following repeated oral administration of red ginseng extract and discuss the possibility of herb-drug interaction based on the unbound plasma concentration of ginsenosides:
(Page 12, line 297-310) Among PPD-type ginsenosides that inhibited in vitro OATP function, ginsenosides Rb1, Rb2, and Rc demonstrated high affinity for OATP1B3 inhibition (1.9 μM – 5.1 μM for OATP1B3; Figure 7). However, the maximum plasma concentrations (Cmax) of Rb1, Rb2, and Rc in rats were in the range of 5.3–15.8 nM following repeated administration of RGE (1.5 g/kg/day) for 7 days (Figure 6) and Cmax of Rb1, Rb2, and Rc in human were 6.2-12.7 nM following repeated administration of RGE (3 g/day) for 14 days [31]. The selected RGE dose in this study is in the range of effective dose without significant toxicity and showed similar plasma concentrations of Rb1, Rb2, and Rc (5.3–15.8 nM in rats and 6.2-12.7 nM in human subjects) [19,33]. In numerous animal studies, the RGE dose has ranged from 200 mg/kg to 2.0 g/kg (i.e., 3–15 mg/kg of total ginsenosides) [36,37]. In human studies, RGE was administered to diabetic patients for 4 to 24 weeks at doses of 2.7 g–6.0g/day, which usually contained 50–100 mg ginsenosides/day [20,38]. The Cmax values following oral administration of ginseng product in both rats and human would be far below the IC50 values required for OATP transport activity inhibition, which contribute to the limited herb-drug interaction between ginseng and OATP/Oatp substrates.

Reviewer 4 Report
In this manuscript, Jeon and colleagues investigate the effects of natural products found in red ginseng extract on various OATP/Oatp transporters. They demonstrate inhibitory effects for some but not all of these compounds in vitro using HEK293 cells overexpressing the human transporters but in vivo, extract given PO at 1.5 g/kg/day for 7 days or one of the more potent OAT inhibitors from RGE, Rc, when given a 3 mg/kg for 5 days IV, failed to effect the PK of valsartan a known OATPB1 and OATPB3 substrate. As a positive control, rifampin, a known inhibitor of both OATBP1 and OATBP3, was shown to effect valsartan PK after just a single dose.
While these are interesting findings, relevant to human health, a number of important controls and experiments remain to be conducted to validate the findings:
Minor
1) Why was such a poor inhibitor of OCT1/OCT2 (TEA) chosen as a control rather than a more potent inhibitor such as atropine or prazosin?
Major
2) While the investigators demonstrated inhibition of uptake of several radiolabeled transporter substrates by the RGE compounds, they did not ever demonstrate their effects in vitro on valsartan uptake. This was conducted for the control rifampin, so the authors appear to have access to instrumentation necessary to measure valsartan accumulation by LCMS. If RGE components are competitive inhibitors, their interaction with the relevant substrate, valsartan, should be measured so that value can be directly compared to in vivo PK values. It is not necessary to do this for all extract components, but perhaps those shown to be most potent against the radiolabeled substrate - Rb1, Rb2, RC.
3) Since the authors are making a concentration argument for the failure of the RGE extract components to effect valsartan PK, they should conversely demonstrate that rifampin concentrations in vivo after the dosing scheme used are above the IC50 values shown in vitro to be effective at inhibiting valsartan accumulation.
4) As indicated in Figure 8, the concentration of Rc in the liver is above 1 uM, which is above the IC50 for Rc for OAT1B3 shown in Fig3. The authors make a protein binding argument, in conjunction with low liver concentrations, to explain the lack of effect of Rc when given IV. To make this argument fully, they should measure protein binding in their in vitro cell culture system and possibly in liver extract such that they can compare free drug levels in both systems.
4) How does the dose of RGE used and in vivo concentrations of extract components achieved in the rat model, compare to usual doses of RGE administered to humans and concentrations of components achieved in man?
Author Response
Responses to Reviewer 4’s comments
In this manuscript, Jeon and colleagues investigate the effects of natural products found in red ginseng extract on various OATP/Oatp transporters. They demonstrate inhibitory effects for some but not all of these compounds in vitro using HEK293 cells overexpressing the human transporters but in vivo, extract given PO at 1.5 g/kg/day for 7 days or one of the more potent OAT inhibitors from RGE, Rc, when given a 3 mg/kg for 5 days IV, failed to effect the PK of valsartan a known OATPB1 and OATPB3 substrate. As a positive control, rifampin, a known inhibitor of both OATBP1 and OATBP3, was shown to effect valsartan PK after just a single dose.
While these are interesting findings, relevant to human health, a number of important controls and experiments remain to be conducted to validate the findings:
Minor
Why was such a poor inhibitor of OCT1/OCT2 (TEA) chosen as a control rather than a more potent inhibitor such as atropine or prazosin?Answer> Since the positive control was selected based on our previous study using the OCT1/OCT2 inhibition using tetraethylammonium and other references [Lee W.K. et al. Organic cation transporters OCT1, 2, and 3 mediate high-affinity transport of the mutagenic vital dye ethidium in the kidney proximal tubule. Am J Physiol Renal Physiol 2009, 296, F1504-1513; Choi M.K. et al. Effects of tetraalkylammonium compounds with different affinities for organic cation transporters on the pharmacokinetics of metformin. Biopharm Drug Dispos 2007, 28, 501-510; Kim S. et al. In Vitro Inhibitory Effects of APINACA on Human Major Cytochrome P450, UDP-Glucuronosyltransferase Enzymes, and Drug Transporters. Molecules 2019, 24]. From the consistent and comparable results of IC50 values, we can conclude the system feasibility for the in vitro inhibition study in HEK293 cells overexpressing drug transporters. However, we agree with the reviewer’s comment on the use of potent inhibitor. In the future study, we’ll use the OCT inhibitors with higher affinity and compare the IC50 values of these inhibitors according to the reviewer’s comments. We ask generous understanding on this issue.
Major
While the investigators demonstrated inhibition of uptake of several radiolabeled transporter substrates by the RGE compounds, they did not ever demonstrate their effects in vitro on valsartan uptake. This was conducted for the control rifampin, so the authors appear to have access to instrumentation necessary to measure valsartan accumulation by LCMS. If RGE components are competitive inhibitors, their interaction with the relevant substrate, valsartan, should be measured so that value can be directly compared to in vivo PK values. It is not necessary to do this for all extract components, but perhaps those shown to be most potent against the radiolabeled substrate - Rb1, Rb2, RC.Answer> As the reviewer suggested, we performed the inhibitory effect of Rb1, Rb2, and Rc on the OATP1B1 and OATP1B3-mediated valsartan uptake and calculated IC50 values. The results and discussion was revised accordingly as follows:
(Page 9, line 214) We further investigated herb-drug interaction between valsartan and individual ginsenoside. At first, the inhibitory effect of ginsenoside Rb1, Rb2, and Rc on the OATP1B1 and OATP1B3-mediated valsartan uptake was measured. Ginsenoside Rb1, Rb2, and Rc was selected considering its stability and high plasma concentation in rat plasma (based on Figure 6) and in human plasma [31,33] as well as its low IC50 value for OATP1B3 inhibition (2.28 μM, 1.76 μM, and 1.36 μM, respectively, Figure 3). As shown in Figure 7, Rb1, Rb2, and Rc inhibited both OATP1B1 and OATP1B3-mediated valsartan uptake in a concentration dependent manner and yielded IC50 values of 8.8-24.1 μM for OATP1B1 and 1.9-5.1 μM for OATP1B3. The results higher affinity of Rb1, Rb2, and Rc to OATP1B3 than OATP1B1 and the lowest IC50 value was shown in Rc inhibition on OATP1B3-mediated uptake of valsartan was consistent with the previous results (Figure 2 and 3).
Since the authors are making a concentration argument for the failure of the RGE extract components to effect valsartan PK, they should conversely demonstrate that rifampin concentrations in vivo after the dosing scheme used are above the IC50 values shown in vitro to be effective at inhibiting valsartan accumulation.
Answer> According to the reviewer’s suggestion, we added reference results that showed rifampin concentration in rat plasma following single oral administrations of rifampin (20 mg/kg) and protein binding results in the revised manuscript as follows:
(Page 12, line 324) In case of co-administration of valsartan and rifampin, a significant drug interaction between valsartan and rifampin was found (Figure 5A) because unbound concentration of rifampin (4.7-22.9 μM) would exceed the IC50 values required for OATP inhibition considering the plasma concentration (over 5 μg/mL for 12 hours and Cmax of 15.7-24.5 μg/mL) and protein binding of rifampin (23.1%) in rats following oral administration of rifampin 20 mg/kg [23,39,40].
As indicated in Figure 8, the concentration of Rc in the liver is above 1 uM, which is above the IC50 for Rc for OAT1B3 shown in Fig3. The authors make a protein binding argument, in conjunction with low liver concentrations, to explain the lack of effect of Rc when given IV. To make this argument fully, they should measure protein binding in their in vitro cell culture system and possibly in liver extract such that they can compare free drug levels in both systems.
Answer> In vitro inhibition study was performed in HBSS (Hank’s balanced buffer solution) which did not contain serum or tissue protein. Therefore, concentrations added in the in vitro inhibition study represent unbound concentration. On the other hand, we agree with the reviewer’s comment on the measurement of protein binding in the plasma and the liver. We measured protein binding of Rb1, Rb2, and Rc in rat plasma and liver homogenates using a rapid equilibrium dialysis kit (ThermoFisher Scientific Korea, Seoul, Korea; molecular weight cut-off 8000 Da). The results and discussion using unbound concentration of Rb1, Rb2, and Rc were revised as follows:
(Page 10, line 257) In addition, these tri-glycosylated ginsenosides showed high protein binding in rat plasma and liver homogenates (Figure 9B). When calculated free Rc concentration in our system, free Rc concentration was estimated to be 0.08-0.34 μM in the rat plasma and 0.07-0.14 μM in the rat liver. As Oatp transporters are located in the sinusoidal membrane of hepatocytes, the low hepatic distribution and high protein binding of Rc may contribute to the negative inhibitory effect of Rc on Oatp transporters in vivo, which might result in the negligible pharmacokinetic interaction between Rc and valsartan. Similarly, limited herb-drug interaction between valsartan and ginsenoside Rb1 and Rb2 would be expected based on their similarity in the structure, protein binding features, and inhibitory effect on OATP transporters (Figures 7 and 9B, Table 1).
(Page 12, line 312) The plasma concentration was ranged from 7.8 μM to 34.1 μM but unbound fraction of tri-glycosylated PPD-type ginsenosides (Rb1, Rb2, and Rc) was very low (0.1-0.2% in rat plasma, 0.4-0.5% in rat liver; Figure 9B). Moreover, the tri-glycosylated ginsenosides are hydrophilic and bulky and, thus, they are difficult to be readily distributed in the liver tissue. Taken together, high protein binding and limited liver distribution of tri-glycosylated PPD-type ginsenosides (Rb1, Rb2, and Rc) might contribute to the lack of in vivo pharmacokinetic herb-drug interactions involving valsartan in rats although their plasma concentration was maximized following repeated intravenous injection of single ginsenoside. Jiang et al. [23] reported that the unbound fraction of PPD-type ginsenosides was very low (0.4–0.9% in Rb1, Rc, and Rd) in the human plasma. Based on the similarity in the structure and protein binding features between rats and human and inhibitory effect on OATP transporters of Rb1, Rb2, and Rc, limited herb-drug interaction between valsartan and ginsenoside Rb1, Rb2, and Rc would be expected in human.
How does the dose of RGE used and in vivo concentrations of extract components achieved in the rat model, compare to usual doses of RGE administered to humans and concentrations of components achieved in man?
Answer> As the reviewer suggested, we compared the plasma concentrations of ginsenosides in rat and haman and added the dose selection criteria in the revised manuscript as follows:
(Page 12, line 299) However, the maximum plasma concentrations (Cmax) of Rb1, Rb2, and Rc in rats were in the range of 5.3–15.8 nM following repeated administration of RGE (1.5 g/kg/day) for 7 days (Figure 6) and Cmax of Rb1, Rb2, and Rc in human were 6.2-12.7 nM following repeated administration of RGE (3 g/day) for 14 days [31]. The selected RGE dose in this study is in the range of effective dose without significant toxicity and showed similar plasma concentrations of Rb1, Rb2, and Rc (5.3–15.8 nM in rats and 6.2-12.7 nM in human subjects) [19,33]. In numerous animal studies, the RGE dose has ranged from 200 mg/kg to 2.0 g/kg (i.e., 3–15 mg/kg of total ginsenosides) [36,37]. In human studies, RGE was administered to diabetic patients for 4 to 24 weeks at doses of 2.7 g–6.0g/day, which usually contained 50–100 mg ginsenosides/day [20,38]. The Cmax values following oral administration of ginseng product in both rats and human would be far below the IC50 values required for OATP transport activity inhibition, which contribute to the limited herb-drug interaction between ginseng and OATP/Oatp substrates.

Round 2
Reviewer 2 Report
The authors did not address the issues raised by the reviewer.
Author Response
Responses to Reviewer 2’s comments – Round 2
The authors did not address the issues raised by the reviewer.
During the second revision, we revised the manuscript according to the reviewer’s Round 1 comments.
(1) The animal handling procedures are not acceptable. Repeated retro-orbital puncture is not acceptable for blood collection.
Answer> As the reviewer pointed, we have repeated retro-orbital puncture for 48 h for the pharmacokinetic study, which is not generally accepted. However, to calculate and compare the pharmacokinetic parameters we performed repeated retro-orbital puncture under isoflurane anesthesia. During the experimental procedure, rats did not suffer from any significant injury or infection. And our animal procedure was approved by the Animal Care and Use Committee of Kyungpook National University (Approval No. KNU 2017-21 and KNU 2019-83).
We added the following description during the revision (Page 14, line 399-404) : All animal procedures were approved by the Animal Care and Use Committee of Kyungpook National University (Approval No. KNU 2017-21 and KNU 2019-83). To calculate and compare the pharmacokinetic parameters of valsartan and ginsenosides, we performed repeated blood sampling through the retro-orbital puncture under isoflurane anesthesia. During the experimental procedure, rats did not suffer from any significant injury or infection.
(2) The study is not clinical relevant. The plasma levels of ginsenosides are so low (figure 6) and they are unlikely cause interaction when judging from their IC50 levels.
Answer> As the reviewer pointed, the lack of herb-drug interaction between red ginseng and valsartan following oral administration of red ginseng extract could be expected because of low plasma concentrations of ginsenosides. However, another purpose of this study was to investigate the inhibitory effect of PPD-type and PPT-type ginsenoside on the transport activity using in vitro cell system (HEK293 cells overexpressing OCT1/2, OAT1/3, OATP1B1/1B3) and to investigate the possibility of herb-drug interaction between Rc (a potent OATP inhibitor among PPD-type and PPT-type ginsenoside tested) and valsartan. We revised the purpose of this study during the revision.
(Page 2, line 84-88) In addition, the inhibitory effect of PPD-type and PPT-type ginsenosides on drug transport activity has not been studied extensively. Considering the growing evidence of herb-drug interactions involving drug transporters [10], the aim of this study was to investigate the effect of RGE and individual ginsenoside (PPD-type as well as PPT-type) on drug transporters using in vitro cell system and/or in vivo animal model.
We also added the lack of herb-drug interaction between red ginseng and valsartan following oral administration of red ginseng extract in both rats and human considering the plasma concentration and IC50 values of ginsenosdies in the Discussion section.
(Page 12, line 302-305) The Cmax values following oral administration of ginseng product in both rats and human would be far below the IC50 values required for OATP transport activity inhibition, which contribute to the limited herb-drug interaction between ginseng and OATP/Oatp substrates.
In addition, we also added the lack of herb-drug interaction beween PPD-type ginsenoside Rc and valsartan and the underlying mechanisms during the revision as follows :
(Page 12, line 311-314) Taken together, high protein binding and limited liver distribution of tri-glycosylated PPD-type ginsenosides (Rb1, Rb2, and Rc) might contribute to the lack of in vivo pharmacokinetic herb-drug interactions involving valsartan in rats although their plasma concentration was maximized following repeated intravenous injection of single ginsenoside.

Reviewer 4 Report
I am satisfied that the authors have addressed the majority of my concerns. My only one remaining suggestion is to ensure that the authors have accurately calculated protein binding in their liver homogenates. They do not report on the calculations employed but I suggest they review the following paper
C. Kalvass and T. S. Maurer. Influence of Nonspecific Brain and Plasma Binding on CNS Exposure: Implications for Rational Drug Discovery. Biopharm. Drug Dispos. 23: 327–338 (2002).which describes a calculation done for tissue binding in the brain that takes into account the dilution of tissue that occurs upon preparation of a homogenate. As applying this correction would only serve to decrease the unbound fraction, it will only strengthen the author's conclusions that the amount of drug available for inhibition of transporters in vivo is too low. However, it will potentially increase the accuracy of the reported values for others. If they have already taken the dilution step into consideration in their reported values, then there is no need for further revision.
Author Response
I am satisfied that the authors have addressed the majority of my concerns. My only one remaining suggestion is to ensure that the authors have accurately calculated protein binding in their liver homogenates. They do not report on the calculations employed but I suggest they review the following paper “C. Kalvass and T. S. Maurer. Influence of Nonspecific Brain and Plasma Binding on CNS Exposure: Implications for Rational Drug Discovery. Biopharm. Drug Dispos. 23: 327–338 (2002).” which describes a calculation done for tissue binding in the brain that takes into account the dilution of tissue that occurs upon preparation of a homogenate. As applying this correction would only serve to decrease the unbound fraction, it will only strengthen the author's conclusions that the amount of drug available for inhibition of transporters in vivo is too low. However, it will potentially increase the accuracy of the reported values for others. If they have already taken the dilution step into consideration in their reported values, then there is no need for further revision.
Answer> Thank you for the valuable comment. We apologize not to describe how to calculate the plasma and liver protein binding. We already calculated the liver tissue binding of Rb1, Rb2, and Rc considering a dilution factor to make 10% liver homogenates. Therefore, the results were not recalculated but we added the equations that we employed during the revision.
(Page 15, line 445-450)
Plasma protein binding was calculated using the following equation [47].
Tissue protein binding was calculated using the following equations, and a dilution factor (D as a value of 10) was used since we used 10% liver homognenates [47, 48].
